# Structure and activation of pro-activin A

Xuelu Wang[1], Gerhard Fischer[1] & Marko Hyvönen[1]

Activins are growth factors with multiple roles in the development and homeostasis. Like all TGF-β family of growth factors, activins are synthesized as large precursors from which mature dimeric growth factors are released proteolytically. Here we have studied the activation of activin A and determined crystal structures of the unprocessed precursor and of the cleaved pro-mature complex. Replacing the natural furin cleavage site with a HRV 3C protease site, we show how the protein gains its bioactivity after proteolysis and is as active as the isolated mature domain. The complex remains associated in conditions used for biochemical analysis with a dissociation constant of 5 nM, but the pro-domain can be actively displaced from the complex by follistatin. Our high-resolution structures of pro-activin A share features seen in the pro-TGF-β1 and pro-BMP-9 structures, but reveal a new oligomeric arrangement, with a domain-swapped, cross-armed conformation for the protomers in the dimeric protein.

---

[1] Department of Biochemistry, University of Cambridge, 80 Tennis Court Road, Cambridge CB2 1GA, UK. Correspondence and requests for materials should be addressed to M.H. (email: mh256@cam.ac.uk).

Activins, members of transforming growth factor β (TGF-β) superfamily, were first isolated from porcine ovarian follicular fluid and identified as activating factors for the release of follicle stimulating hormone[1]. Different additional roles have since been identified for these proteins, including regulation of embryogenesis, development of the reproductive system, wound healing, stem cell differentiation and regulation of immune response[2–5].

Activins are disulfide-linked homo- and heterodimers of four inhibin β chains. Activins A and B (homodimers formed from inhibin $\beta_A$ and $\beta_B$ chains respectively) are the best characterized, while the functions of inhibin $\beta_C$ and $\beta_E$ are poorly understood[6,7]. Activins function both in autocrine and paracrine manner. They conduce signalling by binding to extracellular domains of type I and type II receptors and allowing the kinase domains of type II receptors to phosphorylate those of type I receptors and initiate the intracellular signalling[8]. Type I receptors in turn phosphorylate Smad family of transcription factors, which translocate to the nucleus and activate appropriate target genes[9]. Activin signalling can be regulated by antagonist follistatin and FSTL-3, both of which interact directly with activins, blocking their interactions with receptors and thus inhibiting the signalling process[10,11].

Activins, like all other members in the TGF-β superfamily, are synthesized as larger precursors. The precursor chain consists of an N-terminal signal peptide, a poorly conserved pro-domain of 250–350 residues and a smaller, but much more conserved mature domain. Two precursor chains are typically disulfide-linked through conserved cysteine residues in the mature domains to form covalent dimers. In the case of TGF-β1, also the pro-domains are cross-linked through cysteines. During the secretory pathway, the precursors are processed by furin-like proteases that cleave the polypeptide after a dibasic recognition sequence between the pro- and mature domains, releasing the mature signalling domain from the covalent linkage to the pro-domain[12]. Mutations of this furin site have demonstrated the necessity of cleavage for activation of the growth factors[13], but there are no examples of analyses of the same protein before and after the proteolytic processing, studying the direct effect of the cleavage on the structure and activities of the protein.

The pro-domains play important roles in the biosynthesis, stabilization, transportation and signalling of the growth factors and they have been shown to be essential for the assembly and secretion of dimeric TGF-β1 and activin A[14]. The conserved hydrophobic motif at the N-terminus of the pro-domains has been identified as the key interface that stabilizes the mature domains during synthesis[15]. Association with pro-domains appears also to extend the *in vivo* half-life time of the mature growth factors[16,17] and the pro-domains have been reported to interact with various extracellular matrix (ECM) molecules, facilitating storage and, in some cases, modulating the activation of the mature growth factors in the ECM[18–20]. A number of disease-causing mutations have also been identified in different pro-domains, highlighting their essential role in the biology of these growth factors[21].

The pro-form of TGF-βs is the best understood example of these proteins. Its structure has shown how the pro-domains dimerize through disulfide linkage and form a latent complex with the mature growth factor[22]. In a cellular context the N-termini of the pro-domains are disulfide-linked to the ECM through latent TGF-β-binding proteins (LTBPs)[23], locking the growth factors into the matrix for later activation. Further stimulation, such as integrin-mediated mechanical force, can release the active growth factor from the stronghold of the pro-domain[24]. But not all pro-domains interact with their mature growth factors with such affinity and bind to ECM like pro-TGF-βs. Pro-activin A has shown signalling activity once it has been proteolytically cleaved[13] and instead of an integrin-binding RGD motif, the pro-domain has a Lys-rich region that can bind to the heparin or heparan sulfate chains of proteoglycans[19]. Whether the pro-domains are anchored in the ECM or circulate either alone or with the mature growth factor after secretion is still not clear.

The two published structures of precursor growth factors, pro-TGF-β1 and pro-BMP-9, show large structural variation in their quaternary structures[22,25]. Combined with the poor conservation of the pro-domains, it is difficult to transfer this structural information to other TGF-β superfamily members. Attempt to model pro-activin A based on the structure of pro-TGF-β1 has failed, while the model built from the structure of pro-BMP-9 lacks the structural details of the N-terminus of the pro-domain[17]. The flexibility of mature activin A complicates the modelling further[10].

We have studied the structure and activation of pro-activin A in fine detail using an engineered version of the protein. This enabled us to activate it proteolytically *in vitro* and study the effect of this activation on the structure and function using biochemical, biophysical and structural techniques. The crystal structures we have determined of pro-activin A provide insights into the variable features of the pro-domains in the TGF-β superfamily and analysis of follistatin binding to the pro-complex provides us with a further understanding of the role of this inhibitor.

## Results

**Production and activation of human pro-activin A.** Since its original isolation from porcine ovarian follicle fluid, activin A has been produced mainly in eukaryotic expression systems[14,26–30]. To facilitate structural studies, we have established a method for the production of recombinant human pro-activin A by refolding from *Escherichia coli* produced material. We have engineered the protein to enable proteolytic activation *in vitro*, by replacing the original furin cleavage site with a HRV 3C protease site and removed two N-terminal cysteine residues, which by analogy to pro-TGF-β1 are likely to interact with the ECM, to avoid interference with native disulfide structure of the rest of the protein. Using an optimized protocol adapted from what we use for mature activin A[10], we were able to refold all variants of pro-activin A efficiently and purify them to homogeneity.

Uncleaved pro-activin A shows the expected molecular weight of a dimeric disulfide-linked protein in non-reduced SDS–polyacrylamide gel electrophoresis (PAGE) and the correct monomeric molecular weight when analysed under reducing conditions (Fig. 1a). Size-exclusion chromatography in combination with multiple-angle light scattering (SEC-MALS) analysis shows that pro-activin A is a single species with the expected molecular weight (91.5 kDa, calculated molecular weight 90.9 kDa) (Fig. 1a). This uncleaved pro-activin A precursor is unable to induce the expression of an activin-inducible luciferase reporter gene in HEK293T cells (Fig. 1c), in agreement with previous findings that cleavage is required for bioactivity[13]. As we had engineered an HRV 3C protease cleavage site into our protein, we could mimic the furin processing of pro-activin A *in vitro*. After the protease treatment, pro-activin A migrated as two distinct bands in SDS–PAGE. The pro-domain migrated as a monomeric protein under both reducing and non-reducing conditions, whereas the mature domain migrated as a disulfide-linked dimer in a non-reduced gel, and converted into monomeric species on reduction. SEC-MALS analysis shows that after the cleavage the pro- and mature domains remain associated with each other, eluting at the same volume as the

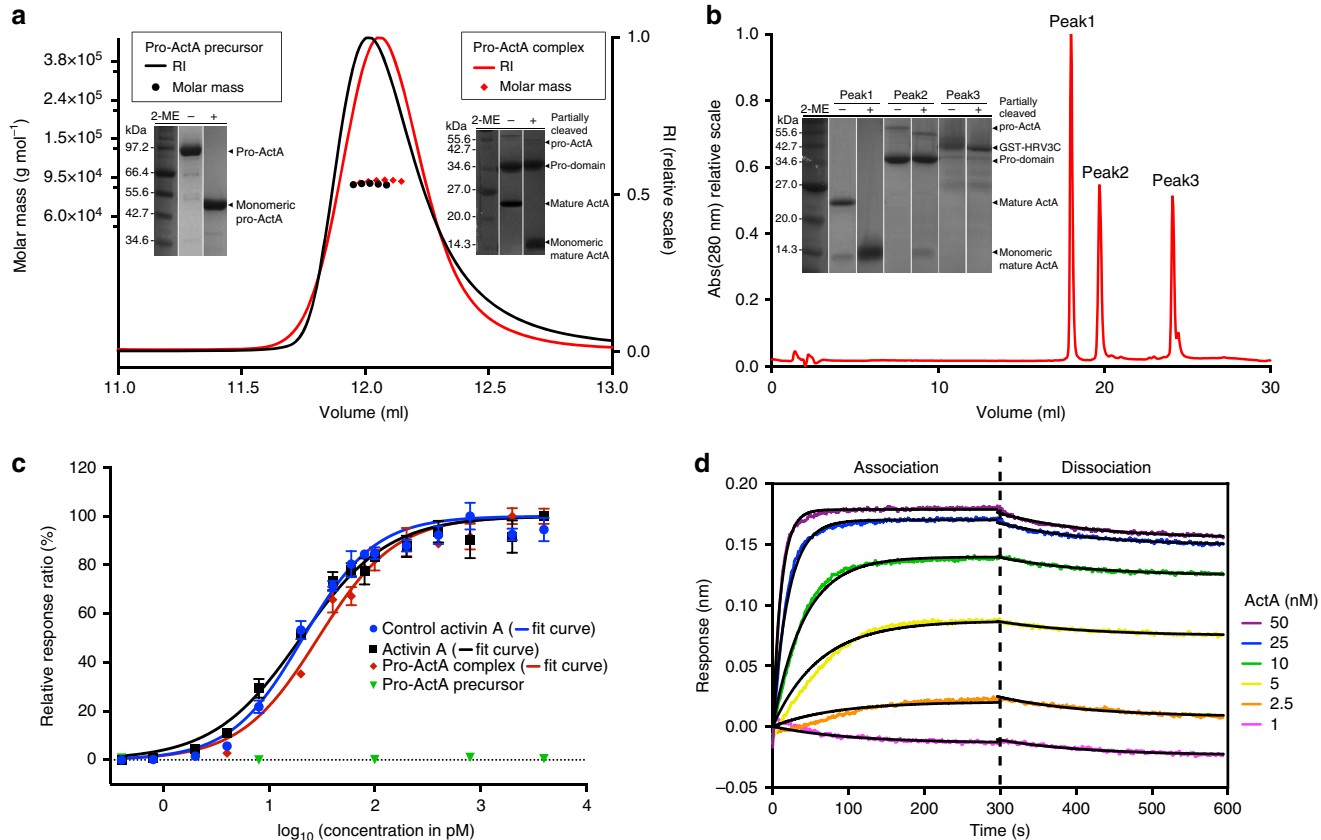

**Figure 1 | Characterization of pro-activin A and its activation. (a)** SEC-MALS analysis of pro-activin A. Size exclusion chromatogram of purified uncleaved pro-activin A (pro-ActA) precursor (black line) and cleaved pro-mature complex (red line) and molecular weight analysis (similarly coloured diamonds) from light scattering for the top of the eluted peak with molar mass scale shown in the left Y axis and refractive index (RI) values in the right Y axis. Peak fractions of each sample are analysed under reducing and non-reducing conditions using SDS–PAGE, shown in the inset (lanes of samples without protein have been removed from the figure). **(b)** Separation of pro- and mature domains of cleaved pro-activin A using reverse phase chromatography. SDS–PAGE analysis of the three peaks is shown in the inset, both under reducing and non-reducing conditions (lanes of samples without protein have been removed from the figure). **(c)** Bioactivity assay of activins. Analysis of bioactivity of control activin A (blue markers and dose response curve), mature activin A purified as in **b** (black), pro-activin A complex (red) and uncleaved pro-activin A precursor (green) using activin A inducible luciferase assay. Data are average of three replicates, normalized to *Renilla* luciferase and scaled to 100% of maximum signal. The error bar shows the s.d. of each data point. **(d)** BLI analysis of pro-domain binding to mature activin A. Coloured curves show the raw data with the concentration of activin A shown at the side of the graph. The black lines show the fitted data.

uncleaved protein and showing dimeric molecular weight of 88.3 kDa (Fig. 1a).

In contrast to the uncleaved precursor, the cleaved pro-activin A complex is highly active in the cell-based luciferase assay with a half-maximal response concentration ($EC_{50}$) of $29 \pm 3$ pM (all the $EC_{50}$ and $IC_{50}$ values and their s.d. values are determined from three replicates). To evaluate the effect of the pro-domain on the bioactivity, the mature domain was purified using reverse phase chromatography, and its bioactivity was compared with an identical protein that was produced without a pro-domain[10] (Fig. 1b). The complex-derived mature activin A conducts signalling marginally better than the pro-mature complex, with an $EC_{50}$ value of $20 \pm 2$ pM, which is in excellent agreement with the activity of our control activin A ($EC_{50} = 21 \pm 2$ pM; Fig. 1c). These data confirm that the refolded and proteolytically cleaved pro-activin A is correctly folded, homogenous and fully biologically active and that the pro-domain has only a marginal effect on the bioactivity.

To understand the activation process of pro-activin A better, we measured the affinity between the pro-domain and the mature activin A using biolayer interferometry (BLI)[31]. The reverse phase purified pro-domain was immobilized onto biosensors using its N-terminal His$_6$-tag and interaction with the mature activin A

was analysed by immersing the sensors into solutions with mature activin A. This allowed us to determine both association and dissociation constants for these binding events and to derive a $K_d$ value of $5.0 \pm 0.2$ nM (all the $K_d$ values and their s.d. values are determined from two replicates; Fig. 1d). We determined the affinity also by immobilizing the pro-mature complex onto the same biosensors, removing mature domain from the complex under non-denaturing conditions and measuring the binding as before. This resulted in a similar dissociation constant of $2.3 \pm 0.1$ nM, in good agreement with the affinity measured using reverse-phase purified pro-domain (Supplementary Fig. 1). With low nanomolar affinity, and considering the $EC_{50}$ of 20–30 pM, it is likely that the pro-activin A complex has mostly dissociated at the concentrations required for full bioactivity. This explains the only marginally lower bioactivity we measured for pro-activin A complex in comparison with mature activin A in our cellular assay. What we do not know is whether *in vivo* the pro-domain associates with other molecules that might stabilize the complex and thus affect the bioavailability of the mature growth factor.

**Crystal structure of pro-activin A.** To understand the atomic details of the interaction between the pro-domain and the mature

activin A, we determined the crystal structures of both the cleaved and uncleaved forms of the protein. While the precursor form described above crystallized, we were unable to increase the diffraction beyond ∼50 Å resolution. To improve the quality of the crystals, we removed residues 259–282 (reported to be part of heparan sulfate-binding site[19]) from the pro-domain. This deletion did not affect folding of the protein and had no effect on the bioactivity of the protein (Supplementary Fig. 2).

The resulting protein yielded significantly better diffracting crystals, enabling us to determine its structure using multi-wavelength anomalous dispersion phasing with selenomethionine-labelled protein. We solved the structure of pro-activin A precursor at 3 Å and refinement of this structure against higher resolution native data yielded a final structure at 2.3 Å. The structure of the cleaved pro-activin A complex was determined at 2.85 Å by molecular replacement using the precursor structure as our starting model. Both structures are highly similar (Supplementary Fig. 3), suggesting that the cleavage does not result in significant changes in the structure of the complex, as has been observed with pro-TGF-β1 (ref. 22). We will therefore use the higher resolution precursor structure in the subsequent analyses.

The structure of the pro-activin A precursor consists of two subunit chains that are disulfide-linked at the mature domains (Fig. 2a). In each subunit, the N terminus of the pro-domain, starting from the first visible residue Q51, forms a helical 'forearm' region consisting of three α-helices (α1, α2 and α3), a latency lasso and a β-hairpin. The structure continues then across the dimeric mature domains, connecting to the globular β-stranded 'shoulder' domain of the pro-domain. The linker between the pro- and mature domains, with the protease cleavage site, is visible neither in the precursor nor in the pro-mature complex. Given the positioning and distance (∼16 Å) between the end of the pro-domain and the start of the mature domain (which is clearly defined), the connectivity shown in Fig. 2a is the only feasible one; the alternative connectivity would require the linker to wrap around the complex and even as a linear distance (36 Å) this is too long for the missing 10 residues to span.

In the structure of pro-activin A, two precursor chains form a cross-armed, domain-swapped dimer. The 'forearm' regions wrap around the mature domains and create a connecting 'bridge' to the globular 'shoulder' parts, which in turn interact with the opposite mature domains (Fig. 2b). The helical bridge connecting the forearm and shoulder is clearly visible in the density map and

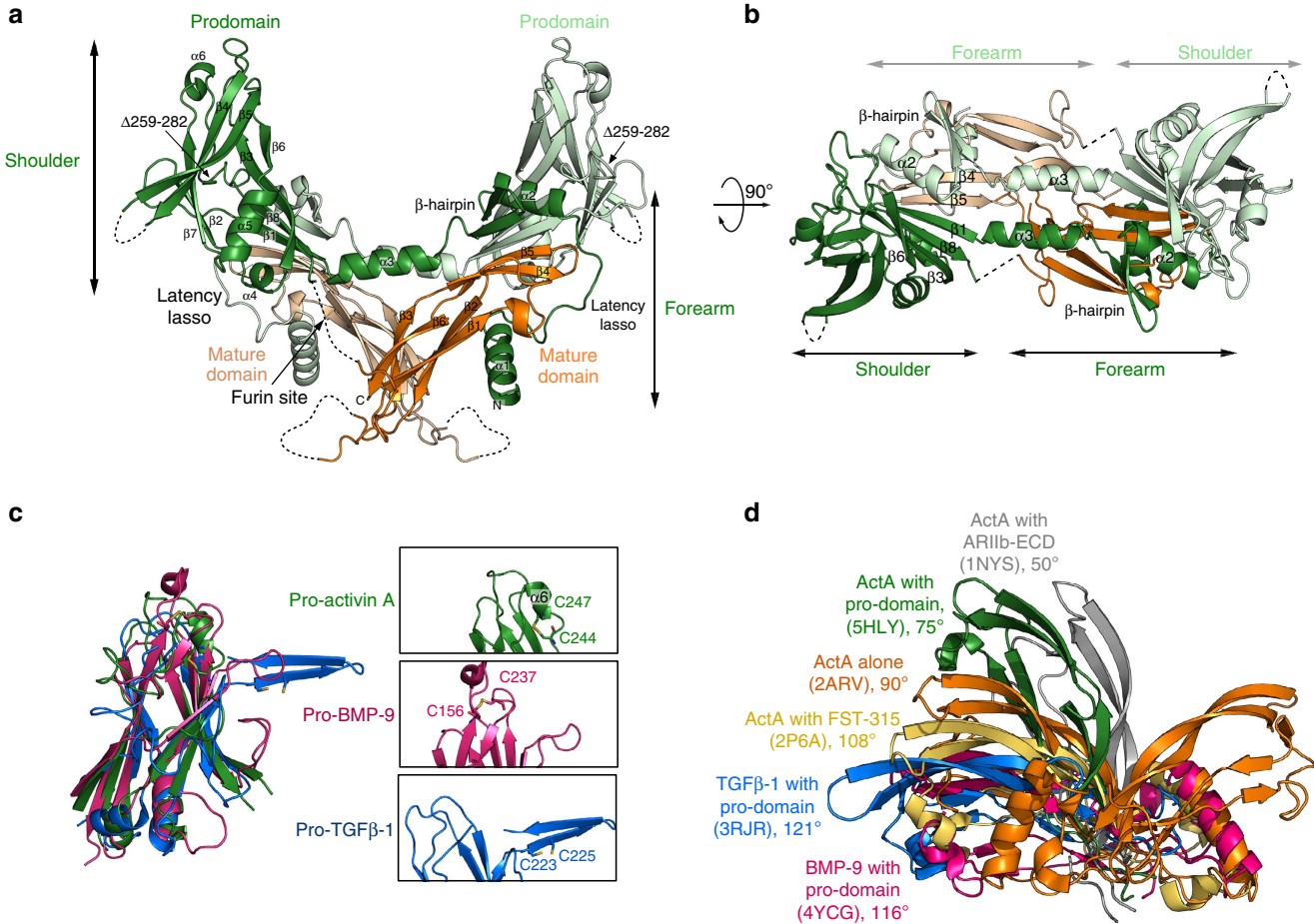

**Figure 2 | Structure of pro-activin A.** (**a**) Quaternary structure of pro-activin A with mature domains coloured in darker and paler orange and pro-domains in green and light green. Different features are labelled in the structure and dotted lines indicate segments that have not been modelled due to unobserved electron density. (**b**) View perpendicular to **a**, along the two-fold axis, showing the β-hairpins and the connecting α-helices that run over the mature domain and create the 'cross armed' organization for the dimer. (**c**) Overlay of shoulder domains of pro-activin A (green), pro-TGF-β1 (blue) and pro-BMP-9 (purple) with insets showing the differing disulfide bonding at the tip of the domain. (**d**) Superpositioning of different mature activin A structures, as labelled in the figure. All the structures are aligned with one protomer only to the apo-structure of mature activin A. The apo-activin A dimer (orange) is shown with both chains and for all the other structures only the unaligned protomer is shown for clarity. The inter-protomer angle is indicated for each dimer, defined as the angle between two Ile100$_m$ residues with the inter-chain disulfide bond as the origin.

the anomalous difference map confirms the position of the selenomethionines in the α3-helix (Supplementary Fig. 5). This organization is in clear contrast to the structures of pro-TGF-β1 (PDB: 3RJR)[22] and pro-BMP-9 (PDB: 4YCG)[25], in which both the forearm and shoulder parts of the pro-domains interact with the mature domain of the other precursor chain in the dimer (see Discussion below).

The shoulder domain of pro-activin A is very similar to those of pro-TGF-β1 and pro-BMP-9, defined by two anti-parallel β-sheets and an α-helix (Fig. 2c). The two cysteines in the pro-domain of activin A form an intra-chain disulfide bond (instead of the inter-chain disulfide bonds found in pro-TGF-β1) connecting the α6-helix with the preceding loop; equivalent disulfide in pro-BMP-9 bridges two neighbouring β-strands.

**Variable conformation of mature activin A.** The structure of mature activin A has been determined a number of times, in isolation[10] and in complex with receptor ecto-domains[32,33], follistatins[34,35] and FSTL-3 (ref. 11). In these structures the mature activin A dimer shows significant variation in its inter-protomer conformation, more than any other TGF-β-like ligand[10]. The inter-protomer angle of dimeric activin A ranges from 50° to 108° (Fig. 2d). Even the most linear, antiparallel conformation for activin A (in complex with FST-315) is still not as open as other ligands, such as TGF-β1, with an inter-protomer angles of 108° and 121°, respectively. TGF-β3 has also been reported to show conformational changes when it is complexed with receptors[36], but to a much lesser extent (115°–125°) compared with activin A.

In the pro-activin A, the mature domain shows more closed conformation (75°) than in most other structures, and much more so than TGF-β1 or BMP-9 in their pro-complexes (Supplementary Fig. 6). Crystal packing could cause some of these variations, but it is clear that activin A is highly malleable and this property has been proposed to play a role in the interaction with its type I receptors[33].

The palm α-helix of the mature domain appears to be disordered in both pro-domain structures, lacking interpretable electron density (Supplementary Fig. 4a). This helix is also missing from the complex of mature activin A with type II receptors where the dimer shows highly closed conformation (50°, PDB: 1NYS)[32]. The closed conformation in both cases causes the connecting loop from the end of the helix (Met68$_m$) to the start of the next β-strand (Cys81$_m$) to pull the helix from the dimer interface. Whether this is also happening in solution, remains to be determined.

**Structure of pro-activin A reveals a new connectivity.** The cross-armed conformation of pro-activin A is different from the closed ring-shaped conformation in pro-TGF-β1 and the widely open conformation in pro-BMP-9 and the connectivity in pro-activin A is different too (Fig. 3a–c)

The structure of pro-activin A shows a continuous polypeptide chain in the protomeric precursor chain with the exception of the loop between β4 and β5 strands, the activation loop and the mature domain α-helix (Supplementary Fig. 4a). Although the electron density for the linker between the pro-domain and the mature domain (Glu301-Arg310) is not visible, it has been confirmed to be intact by SDS–PAGE analysis of dissolved crystals (Supplementary Fig. 4b). One of the key features of the oligomeric architecture is the connection between the forearm region and the shoulder domain. Unambiguous electron density shows how the pro-activin A forearm region is connected by a long α-helix to the shoulder domain that sits on the top of the opposite mature domain (Fig. 3a). In contrast, the structure of pro-TGF-β1 appears to be in a different oligomeric arrangement

with the forearm and shoulder of one pro-domain interacting with the mature domain from the other chain[22] (Fig. 3b). However, the electron density for the loops (Glu62-Pro68) between forearm and shoulder region are not visible in the pro-TGF-β1 structure and the connectivity cannot be unambiguously defined. Given the similarity of the pro-activin A structure, it is tempting to speculate that pro-TGF-β1 has in fact the same cross-armed structure as pro-activin A. The distances between Glu62 and Pro68 in the pro-TGF-β1 structure are ∼15 Å if connected as in the original structure and ∼17 Å if the connectivity would be the same as for pro-activin A (Supplementary Fig. 7a,b), and both arrangements are therefore possible. In the structure of pro-BMP-9, the end of the α2-helix is very close to the start of the shoulder part and it could not be connected in the same way as pro-activin A is.

**Implications on other TGF-β precursors.** Despite the highly divergent overall conformations, the protomeric subunits of pro-activin A, pro-TGF-β1, and pro-BMP-9 share many features and comparison of these structures allows us to predict some common structural features in the TGF-β superfamily.

The α1- and α2-helices and the latency lasso in the forearm region of pro-activin A form a hydrophobic interface with the mature domain (Fig. 3d), as in the structure of pro-TGF-β1 (Fig. 3e). A notable difference between these two structures is the disorder of the α-helix from the palm region of the mature domain in activin A, with the α1-helix of the pro-domain is occupying this position. The α2-helix of pro-activin A has a slightly different orientation compared with that of pro-TGF-β1 and makes hydrophobic contacts with the following β-hairpin (which is not present in pro-TGF-β1). Despite these differences, the hydrophobic residues in the forearm that define the interface with the mature domain are well conserved between pro-activin A and pro-TGF-β1. In the structure of pro-BMP-9, the α1-helix is replaced by α5-helix in the concave site of the mature domain and therefore cannot be compared directly. Structure-based sequence alignment of the selected members of the TGF-β superfamily reveals a conserved hydrophobic motif in the forearm region (Supplementary Fig. 8). High sequence similarity in the regions of α1- and α2-helices and the fully conserved leucine and proline residues in the latency lasso suggest a common interface between the pro-domain and the mature domain in different members of the TGF-β family.

The core of the globular shoulder domain of the pro-domain is also well conserved between the three structures (Fig. 2c). One of the β-sheets extends to the mature domain, forming an extended β-sheet in pro-activin A and pro-BMP-9 structures. In pro-activin A, this extended β-sheet is further strengthened by Glu139, Ser141 and Glu143 from the shoulder domain making polar contacts with Asp95$_m$, Asn99$_m$ and Lys103$_m$ from the mature domain (Fig. 3g). A similar extended β-sheet presents in the structure of pro-BMP-9, but it lacks the additional polar contacts and the interaction between the β-strands in the pro-domain and the mature domain appears less intimate (Fig. 3i). In pro-TGF-β1, the β-sheet of the pro-domain is rotated about 90° from the mature domain and adopts a vertical conformation, constrained by the inter-chain disulfide bond at the bowtie regions (Fig. 3h).

**Regulation of pro-activin A by follistatin.** Follistatin (FST) is a well-studied antagonist of activins and the mechanism by which it inhibits activin A signalling has been well characterized[10,34]. Its ability to inhibit pro-activin A is less well known and we have therefore analysed the ability of FST to interact with pro-activin A. Follistatin is found in two main isoforms, follistatin-288 (FST-288) and follistatin-315 (FST-315), as a result of alternative

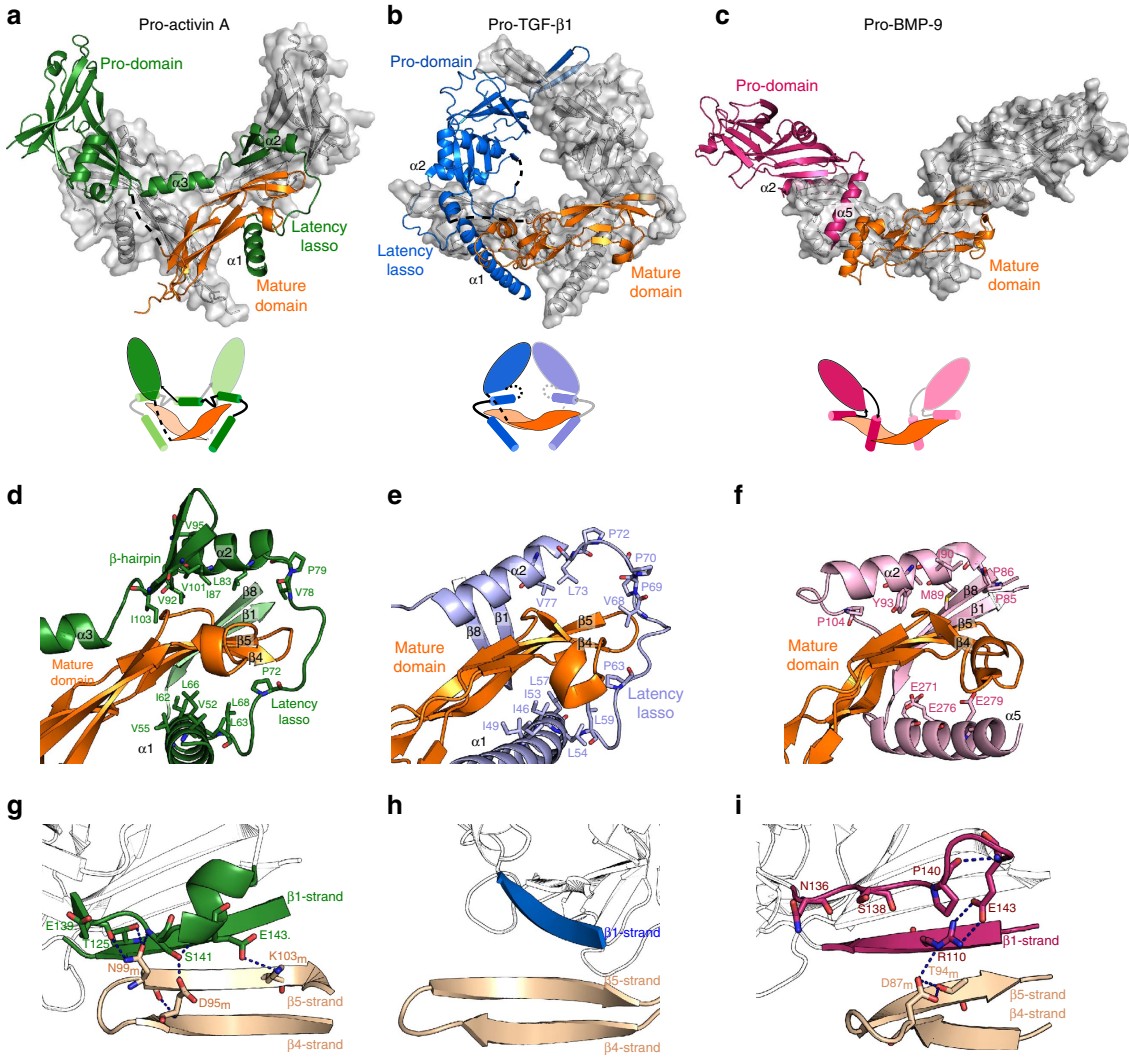

**Figure 3 | Comparison of pro-activin A with pro-TGF-β1 and pro-BMP-9.** (**a**–**c**) Structures of pro-activin A (green pro-domains), pro-TGF-β1 (blue) and pro-BMP-9 (purple) shown perpendicular to the vertical two-fold axes, highlighting the different conformations of the three structures with one protomer chain in each structure shown with surface. In all cases the mature domains are coloured with darker and lighter shades of orange and dotted lines depict the missing parts of the models. Simplified diagrams underneath illustrate the different connectivites for the complexes. (**d**–**f**) Detailed views of the association of the N-terminal segment of the pro-domain with the mature domain in pro-activin A (green and orange), pro-TGF-β1 (light blue and light orange) and pro-BMP-9 (pink and light orange). (**g**–**i**) Interaction of the mature domain finger with the β-sheet of the pro-domain. Colouring as in the figures above, with dotted lines indicating side chain-mediated hydrogen bonds between the polypeptides.

splicing of the last exon[37], both of which can inhibit activin signalling[34,35]. FST-288 has been reported to be more efficient in the inhibition of activin A signalling with a $K_d$ of $46.5 \pm 0.37$ pM (ref. 38) and we have therefore used this shorter isoform in our studies.

In the luciferase-based bioassay, with pro-activin A complex and mature activin A at the concentration of 60 pM, FST-288 inhibited the signalling of both the pro-mature complex and the mature dimer, with IC$_{50}$ values of $90 \pm 9$ and $81 \pm 3$ pM, respectively (Fig. 4a). This demonstrates that the pro-domains do not seem to affect the ability of FST-288 to inhibit activin A. To understand whether FST-288 forms a complex with the pro-domain-associated activin A or with the mature domain only, we analysed activin–follistatin interaction by co-immunoprecipitation with 200 nM pro-activin A complex (40× above the dissociation constant of 5 nM) and 1.2 μM FST-288, capturing the pro-domain with its N-terminal His$_6$-tag (Fig. 4b). Without FST-288, the mature domain activin A can be co-precipitated with the pro-domain, as expected. When FST-288 was mixed

with the pro-activin A complex, only the pro-domain was observed in the precipitant, suggesting FST-288 binds only to the mature activin A and competes with the pro-domain. In a reverse experiment we used an anti-FST-288 antibody to precipitate FST-288 and any proteins bound to it. In this case, the pro-domain remained in solution and only the mature protein was precipitated with FST-288 (Fig. 4c), confirming that FST-288 binds only to the mature activin A and displaces the pro-domain from its complex with the mature domain.

As shown earlier, the binding affinity between the pro- and mature domain is relatively modest at 5 nM. The displacement of pro-domain by FST-288 could thus be due to the dissociation of the pro-activin A complex, with FST-288 binding to the released mature domain, rather than through active competition with pro-domain. To distinguish between these two models, we measured the dissociation rate of the mature activin A from the pro-domain in the presence and absence of FST-288 (Fig. 4d). His$_6$-tagged pro-activin A complex was immobilized on sensor tips through the pro-domain at different concentrations and then

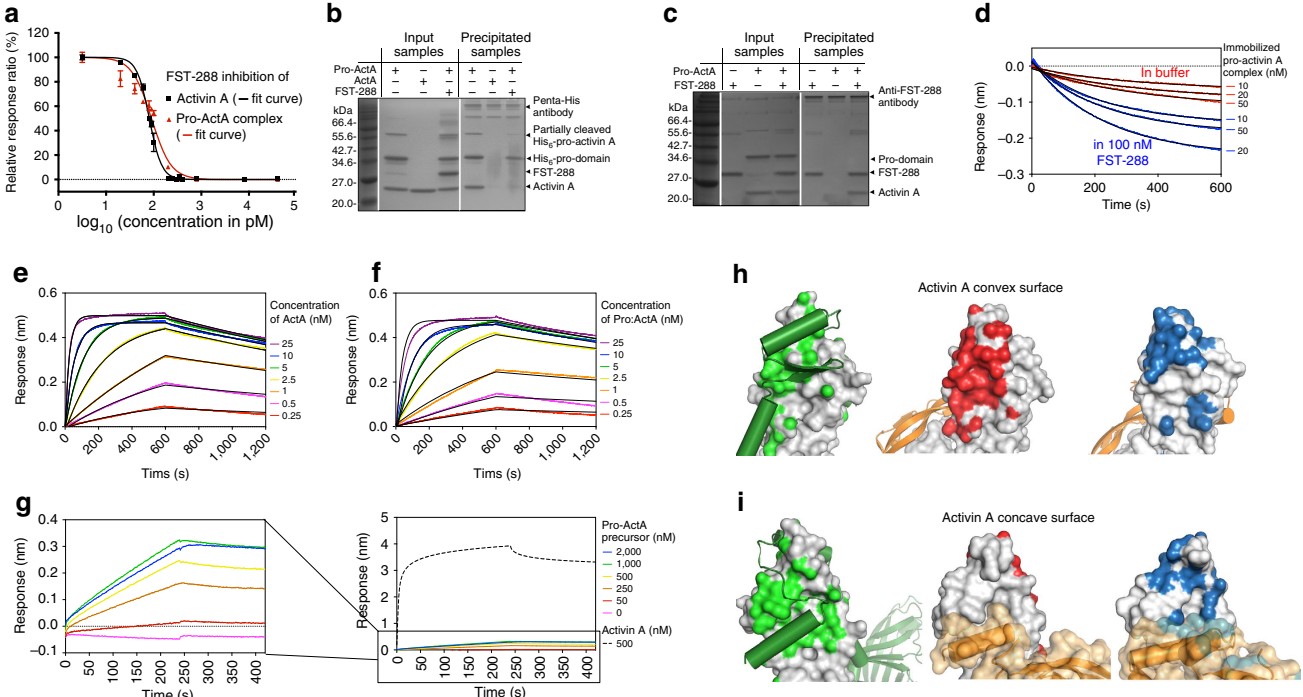

**Figure 4 | Activin–follistatin interaction. (a)** Inhibition of activin A signalling in luciferase-based bioassay using increasing concentrations of FST-288 with IC$_{50}$ curve for each assay as solid line. Data for mature activin A are shown in black and data for pro-activin A complex in red. Data is the average of three replicates, normalized to *Renilla* luciferase and scaled to 100% of maximum signal. The error bars show the s.d. of each data point. **(b)** Non-reducing SDS–PAGE analysis of immunoprecipitation of FST-288 using anti-penta-His antibody binding to His$_6$-tagged pro-activin A with samples before precipitation on the left and precipitated samples on the right **(c)** Non-reducing SDS–PAGE analysis of immunoprecipitation of activin A using anti-FST antibody binding to FST-288 with samples before precipitation on the left side and precipitated samples on the right. **(d)** Dissociation of mature activin A from pro-mature complex in the presence (blue points) and absence (red points) of FST-288 in the dissociation buffer, with fitted dissociation curves as black lines. **(e)** BLI analysis of immobilized FST-288 binding to mature activin A. Coloured curves show the raw sensogram data with fitted data as black lines. **(f)** BLI analysis of immobilized FST-288 binding to cleaved pro-activin A complex. Colouring as in **e**. **(g)** Sensograms for immobilized FST-288 binding to uncleaved pro-activin A precursor, with pro-activin A precursor concentrations indicated on the far right. The graph on the left shows a zoomed-in view of the data in the right panel in which the binding data for 500 nM mature activin A is shown as a black dashed line for reference. **(h,i)** Interactions of mature activin A on the convex type II receptor-binding surface **(h)** and the concave putative type I receptor-binding surface **(i)**. Left panels show the pro-domain in the pro-mature precursor (green cartoon) binding to mature activin A (shown as surface) with the interaction surface coloured green. The middle panels shows the type II receptor-interacting surface in red (PDB:1S4Y) and the rightmost panels show interactions with follistatin in blue (PDB: 2B0U). The second protomer of mature activin A is shown as orange cartoon.

immersed either into buffer alone or into buffer containing 100 nM FST-288, monitoring the dissociation of the mature domain in both cases. The dissociation was clearly faster in the presence of FST-288 for all levels of immobilized pro-activin A complex (Fig. 4d), with the dissociation rate increasing from 2.99 ($\pm 0.02$) $\times 10^{-3}$ to 4.45 ($\pm 0.02$) $\times 10^{-3}$ s$^{-1}$ (these dissociation rates and their s.d. values are determined from two replicates), indicating that FST-288 can actively displace the pro-domain and extract mature activin A from its grip. This suggests that follistatin can exert its inhibitory effect on activin A even when the mature domain is complexed with its pro-domain, by dissociating this complex and binding to the mature dimer. Given the binding mode of follistatin, where one follistatin interacts with one side of activin A dimer, we speculate that dissociation of one N-terminal pro-domain epitope on the mature activin A could allow follistatin to compete for this site and prevent re-binding, and thereby promote dissociation of the second pro-domain and complete separation of pro- and mature domains. Follistatin binds also to myostatin, but with a slightly lower affinity (12.3 nM) (ref. 39). The affinity between myostatin pro- and mature domains is very similar (8.3 nM) to what we have determined for pro-activin A complex[20] and we would predict that follistatin can act on myostatin pro-mature complex in a similar way as it does on activin A.

As uncleaved pro-activin A is biologically inactive, it was not possible to analyse its interaction with FST-288 using the luciferase-based bioassay and we characterized this interaction using BLI. Avi-tagged FST-288 was immobilized on streptavidin biosensors and interactions were analysed by monitoring binding to different forms of activin A (Fig. 4e–g). The response curves of FST-288 binding to the mature activin A and cleaved pro-activin A complex reached the equilibrium very quickly at 25 nM analyte concentration, resulting in very similar $K_d$ values of 209.0 $\pm$ 0.6 and 258.6 $\pm$ 1.2 pM, respectively. Similarly to the bioactivity assays, most of the pro-domains will have dissociated from the mature domains, with pro-domains having only modest inhibitory effect on FST-288 binding. The association rate of FST-288 binding to pro-activin A precursor, however, was very slow and the response did not reach the plateau in the association phase even at 2 μM analyte concentration, suggesting that the covalent linkage between the pro-domain and mature domain in pro-activin A precursor hinders FST-288 binding and prevents it from displacing the pro-domain from the mature domain.

**The pro-domain covers receptor and follistatin binding sites.** The pro-domain has been shown to have an inhibitory effect on activin A signalling at high concentrations, with an IC$_{50}$ value around 100 nM (refs 15,28). The structure of pro-activin A

provides support to these findings. It shows that the receptor-binding surfaces of activin A are covered by the pro-domain (Fig. 4h,i). The hydrophobic residues (Leu83, Ile87, Leu90, Val92, Ile103 and Ile107) on the forearm region of the pro-domain interact with the residues in the type II receptor-binding epitope (Fig. 3d). It is apparent that binding of the pro-activin A complex to the type II receptor requires displacement of the pro-domain, consistent with the findings in previous studies[15]. Since the covalent linkage between the pro-domain and the mature domain is not in the way of the type II receptor-binding site, the inability of the pro-activin A precursor to induce signalling suggests the linkage may hinder type I receptor binding and hence inhibit signalling.

Similarly, all the key FST-288 binding sites are also occupied by the pro-domain (Fig. 4h,i). The forearm region of the pro-domain covers the convex surface of the mature activin A that binds to follistatin domain 1 and 2 (FS12) and the α1-helix of the pro-domain occupies the concave surface of the mature activin A that binds the α-helix of follistatin N-terminal domain. The groove between the two fingertips of activin A is occupied by the pro-domain latency lasso and the side chain of Arg71 forms a salt bridge with $Asp27_m$ of the mature activin A, which in turn makes polar contact with Arg192 in complex with follistatin. The fingertips of activin A also adopt a different orientation due to formation of the extended β-sheet with the pro-domain.

## Discussion

We have shown using highly purified pro-activin A how the cleavage between the pro- and mature domains fails to dissociate the complex at moderate concentrations, yet it renders the growth factor fully active in our bioassays. In our previous research, we found mature activin A is poorly soluble in physiological buffers and tends to aggregate readily[10]. In contrast, the pro-activin A complex is highly soluble, and amenable to purification in native conditions. It is therefore conceivable that one of the biological roles for the pro-domain is to facilitate the solubility and in vivo distribution of the mature growth factor, reducing unwanted interactions and delivering the protein to the site of action. Activins and BMPs act as morphogens in developing embryos. Diffusion of these proteins in the embryonic tissues has been studied, but mainly using mature domains only and by detecting their activity in the responsive cells[40]. The role of the pro-domain is less well understood, even ignored, and more work is needed to understand this fully.

It has been suggested that the pro-domains promote the synthesis and assembly of mature growth factors, acting as molecular chaperones[14]. These claims usually stem from the notion that deletion of the pro-domain prevents the mature domain from being secreted from the cells. There are many reasons why such deletion could affect secretion of the mature domain, and direct evidence for the chaperone function is missing. With our new structures of pro-activin A, and the revised connectivity of the complex, the chaperone function seems more compelling (Supplementary Fig. 9). The forearm region with the latency lasso could be stabilizing the mature domain of the same polypeptide in a conformation that is compatible with dimerization, while protecting the largely hydrophobic surfaces that bind the receptors. This monomeric protein with its uncomplexed globular shoulder domain is then free to find another monomer in the endoplasmic reticulum to dimerize with. The shoulder domain can interact with the fingers of the opposing mature domain, which are in turn protected by its own forearm epitopes. This mechanism would also be well posed to facilitate dimerization of the growth factors, as individual chains are able to fold more completely and obtain a

conformation that is compatible with dimerization. The connectivity and interactions between the pro- and mature domains could determine dimerization propensity as only mutually compatible monomeric pro-forms could interact with each other in either homo- or heterodimeric manner.

The role of pro-domain in confirming latency to the TGF-βs is well understood and this is largely due to the additional interaction between the pro-domain and the ECM. There is significant evidence that other pro-domains also interact with the components of the ECM, and activin A pro-domain is known to bind to heparan sulfates. These interactions can act as additional glue that keep the pro-domains together, or immobilize them to a desired location in vivo, and thus affect the activity of the growth factors. Intriguingly, like TGF-βs, activins, myostatin (GDF8) and GDF11 contain additional cysteines in the N-termini of their pro-domains and these are likely places for immobilization to the ECM, or for interactions with other proteins. Myostatin is thought to interact with LTBP1, but no data are available for activin A. Johnson et al.[17] have recently shown that pro-activin A is secreted from HEK293F cells as soluble complex, but identification of stoichiometric disulfide-bonded partners would require the components to be expressed at similar levels, and understanding these interactions will require the analysis of proteins in their natural context and in physiological concentrations, as has been done with latent-TGF-β (refs 41,42).

In addition to understanding the physiology of these growth factors, the pro-domain complexes offer great opportunities for engineering new functions to these growth factors and to regulate them in the extracellular milieu. A chimeric pro-domain consisting of the N-terminal region of activin A pro-domain and the C-terminal region of TGF-β1 pro-domain has been used as an inhibitor for activin A[28], reflecting the shared mechanism that these proteins interact with the mature domains, and the different levels of regulation/latency they confer to the mature growth factor. Undoubtedly more such engineering will be inspired by the analysis of these structures.

## Methods

**Pro-activin A constructs and protein expression.** The construct of human inhibin βA chain (residues 30–426, Uniprot P08476) was cloned by PCR into pHAT2 expression plasmid, with resulting fusion gene encoding the subunit of human pro-activin A with an N-terminal $His_6$ tag. Mutagenesis to introduce the HRV 3C protease site and to remove the lysine-rich loop was performed by two-step PCR protocol using overlapping oligonucleotide across the mutated region; all oligonucleotides used for cloning are listed in the Supplementary Table 1.

The expression constructs were transformed into competent cells BL21(DE3) + pUBS520, grown overnight at 37 °C on LB-agar plates containing 100 µg ml$^{-1}$ of ampicillin and 25 µg ml$^{-1}$ kanamycin. Large-scale expression was performed by inoculating 1 litre of 2YT media using colonies collected from LB-agar plate that has been grown overnight at 37 °C. Protein expression was induced by 400 µM IPTG at $OD_{600}$ of 0.6–0.8 and carried out at 37 °C for 4 h. Cells were then pelleted by centrifugation at 5,020g for 20 min and re-suspended in MilliQ water and stored at −20 °C.

**Protein refolding and purification.** The cell pellet from 1 liter of bacterial culture was suspended in lysis buffer (50 mM Tris-HCl pH 8.0, 0.5 mM EDTA, 10 mM DTT) with 0.5% (v/v) Triton-X100 and lysed using the Emulsiflex C5 homogeniser. The lysate was incubated with DNase I and 4 mM $MgCl_2$ for 20 min at room temperature before it was centrifuged at 15,000g for 20 min to pellet the insoluble inclusion bodies. The inclusion bodies were washed in lysis buffer with 0.5% Triton-X100, followed with lysis buffer with 1 M NaCl, and then with lysis buffer only. The washed inclusion bodies were re-suspended in 5 ml of 100 mM neutralized tris(2-carboxyethyl)phosphine (TCEP) and proteins were denatured by addition of 15 ml of solubilization buffer (8 M guanidine hydrochloride (GndHCl), 50 mM Tris-HCl pH 8.0, 10 mM EDTA, 0.1 M cystine), followed by 1-h incubation at room temperature. Denatured solubilized pro-activin A sample was clarified by centrifugation at 15,000g for 20 min. Before refolding, the denatured protein was buffer exchanged into 6 M urea and 20 mM hydrochloric acid, adjusted to 1 mg ml$^{-1}$ and then diluted to the concentration of 0.1 mg ml$^{-1}$ into cold

**Table 1 | Crystallographic data collection and structure determination statistics.**

| | Pro-activin A complex PDB:5HLZ | Pro-activin A precursor PDB:5HLY | SeMet pro-activin A precursor | | |
|---|---|---|---|---|---|
| *Data collection* | | | | | |
| Space group | $P3_1$ | $P6_1\,2\,2$ | $P6_1\,2\,2$ | | |
| Cell dimensions | | | | | |
| $a, b, c$ (Å) | 47.2, 47.2, 445.4 | 46.6, 46.6, 444.9 | 47.0, 47.0, 448.1 | | |
| $\alpha, \beta, \gamma$ (°) | 90.0, 90.0, 120.0 | 90.0, 90.0 120.0 | 90.0, 90.0, 120.0 | | |
| | | | Peak | Inflection | Remote |
| Wavelength | 0.97949 | 0.97866 | 0.97949 | 0.97934 | 0.97204 |
| Resolution (Å) | 44.5-2.85 (3.01-2.85)* | 74.15-2.30 (2.38-2.30)* | 37.05-2.90 (2.91-2.90)* | 37.05-2.90 (2.91-2.90)* | 37.05-2.91 (2.92-2.91)* |
| $R_{sym}$ or $R_{merge}$ | 0.114 (0.948) | 0.107 (0.687) | 0.109 (0.575) | 0.116 (0.627) | 0.121 (0.549) |
| $I/\sigma I$ | 9.7 (1.6) | 17.3 (2.4) | 17.5 (1.4) | 17.3 (1.5) | 16.1 (1.7) |
| Completeness (%) | 99.9 (99.8) | 79.3 (37.1) | 83.7 (19.8) | 84.0 (20.7) | 83.0 (19.2) |
| Redundancy | 5.1 (4.9) | 14.3 (7.3) | 10.9 (2.5) | 10.9 (2.5) | 11.0 (3.3) |
| *Refinement* | | | | | |
| Resolution (Å) | 40.90-2.85 | 74.15-2.30 | | | |
| No. of reflections | 25,878 | 18,900 | | | |
| $R_{work}/R_{free}$ | 0.220/0.273 | 0.2025/0.2524 | | | |
| No. of atoms | | | | | |
| Protein | 9,153 | 2,380 | | | |
| Ligand/ion | 0 | 1 | | | |
| Water | 10 | 25 | | | |
| *B*-factors (Å$^2$) | | | | | |
| Protein | 69.6 | 44.2 | | | |
| Ligand/ion | NA | 59.4 | | | |
| Water | 49.3 | 35.4 | | | |
| r.m.s. deviations | | | | | |
| Bond lengths (Å) | 0.004 | 0.003 | | | |
| Bond angles (°) | 0.810 | 0.724 | | | |

*For each structure, a single crystal was used. Values in parentheses are for the highest resolution shell. NA, not applicable.

refolding buffer (100 mM Tris-HCl pH 8.0, 100 mM ethanolamine pH 8.0, 1 M pyridinium propyl sulfobetaine, 0.5 mM EDTA, 0.2 mM cystine and 2 mM cysteine) with vigorously stirring. The refolding process was allowed to proceed for 7 days at 4 °C.

Refolded protein solution was adjusted to pH 6.0 and was then loaded onto the HiTrap SP HP 5 ml column (GE Healthcare) that had been equilibrated with 50 mM MES pH 6.0. The bound sample eluted with a linear gradient to 50 mM MES pH 6.0, 1 M NaCl in 20 column volumes. Peak fractions with dimeric pro-activin A were pooled and purified further by size exclusion chromatography using HiLoad 16/600 Superdex 200 prep grade column (GE Healthcare) in 10 mM Tris pH 8.0, 500 mM NaCl and 0.5 mM EDTA pH 8.0. Reduced and non-reduced SDS–PAGE was used to analyse the purified proteins.

Selenomethionine-labelled protein was expressed in minimal medium using metabolic suppression method to minimized endogenous methionine production and refolded and purified like the unlabelled protein. Selenomethionine incorporation was confirmed to be complete by mass spectrometry.

For the production of cleaved pro-activin A complex, engineered pro-activin A with its natural furin cleavage site replaced by HRV 3C cleavage site was mixed with recombinant GST fused HRV 3C protease at a mass ratio of 5 to 1 and incubated at 4 °C for overnight. The cleaved pro-activin A was separated from GST-tagged protease by passing the mixture through Glutathione Sepharose 4 Fast Flow beads (GE Healthcare).

**SEC-MALS analysis.** SEC-MALS analysis was performed using a Superdex 200 Increase 10/300 column (GE Healthcare) connected to the DAWN HELEOS II light scattering detector (Wyatt Technology) and the Optilab T-rEX refractive index detector (Wyatt Technology). Bovine serum albumin (Thermo Scientific) was used to calibrate the system in the elution buffer (10 mM Tris pH 8.0, 500 mM NaCl) before 100 µl of protein sample at the concentration of 1.5 mg ml$^{-1}$ was analysed. The experimental data were recorded and processed by ASTRA (Wyatt Technology) software.

**Co-immunoprecipitation.** Pro-activin A complex and its mixture with follistatin-288 (FST-288) were analysed by immunoprecipitation (IC) assay using Protein G magnetic beads (ThermoFisher). A volume of 50 µl of the magnetic beads was incubated with 10 µg of penta-His antibody (Qiagen, catalogue no. 34660) to capture the His$_6$-tagged pro-domain of pro-activin A complex. The antibody-bound magnetic beads were washed three times using phosphate-buffered saline (PBS) and split into three tubes to be incubated with 200 nM pro-activin A

complex, 200 nM activin A or the mixture of 200 nM pro-activin A complex with 1.2 µM FST-288, respectively, for 30 min at room temperature. Similar experiment was performed using magnetic beads that were immobilized with anti-FST-288 antibody (Ansh Labs LLC, catalogue no. AB-307-AF002) at the same bead/antibody ratio. Three samples, 200 nM pro-activin A complex, 600 nM FST-288 or the mixture of 200 nM pro-activin A complex with 600 nM FST-288 were mixed with the antibody-bound magnetic beads. PBS with additional 400 mM NaCl was used to wash these magnetic beads three times to eliminate non-specific bound proteins. The bound proteins were then eluted from the magnetic beads by 10 mM glycine pH 1.7 and analysed by SDS–PAGE. Uncropped figures of the SDS–PAGE analyses are shown in the Supplementary Fig. 10.

**Luciferase assay.** To analyse the signalling activity of activin A in different protein forms, a cell-based luciferase assay was established. HEK293T cells (ATCC, catalogue no. CRL-3216; a generous gift from Dr Trevor Littlewood, Department of Biochemistry, University of Cambridge) were cultured in 96-well flat-bottom cell culture plates using Dulbecco's Modified Eagle Medium (DMEM; Life Technologies) with 10% (v/v) fetal bovine serum (FBS; Life Technologies) at 37 °C in a humidified incubator with 5% CO$_2$. When the confluence of cells reached 80%, 33 ng of pGL3-CAGA (carrying activin A responsive firefly luciferase gene) and 17 ng of pRL-SV40 (Promega, with constitutively expressed *Renilla* luciferase) plasmids were transfected into the cells in each well using 0.2 µl of FuGENE HD transfection reagent (Promega). After overnight incubation, the cells were washed with sterile PBS and cultured in DMEM with 0.5% FBS. Serial dilutions of activin A and its pro-forms in DMEM with 0.5% FBS were added into the cell culture and the experiments were performed in triplicate. In the FST-288 inhibition assay, FST-288 was diluted in DMEM containing 0.5% FBS and 60 pM activin A or pro-activin A. The mixture samples were then added into the cell culture in triplicate. After overnight incubation, cells were washed with PBS and lysed using 20 µl of passive lysis buffer (Promega). A volume of 5 µl of cell lysate in each well was transformed into Corning 96-well flat-bottom white plate (Sigma) and was mixed with 25 µl of Dual-Glo Luciferase Reagent (Promega). After incubation with shaking for 30 min, the firefly luciferase was measured using PHERstar microplate reader (BMG LABTECH). A volume of 25 µl of Dual-Glo Stop & Glo Reagent (Promega) was then added into each well to quench the signal from firefly luciferase and to provide substrate for the *Renilla* luciferase. The luminescence was measured using BMG PHERAstar microplate reader. The final response was normalized using the firefly luminescent signal divided by the *Renilla* luminescent signal. Relative response ratio was calculated as $(R - R_{min})/(R_{max} - R_{min})$ to allow numerical comparison

between assays from different plates, where $R_{min}$ is the minimal response from the negative control and $R_{max}$ is the maximal response from the positive control in each plate.

**Biolayer interferometry.** To analyse protein interactions, biolayer interferometry experiments were performed using Octet RED96 (ForteBio) instrument at 30 °C. The interactions between pro- and mature domain of activin A were analysed using anti-penta-HIS biosensors immobilized with N-terminal His$_6$-tagged pro-domain or pro-activin A complex. The biosensors immobilized with pro-activin A complex were dipped into blank buffer or buffer containing 100 nM FST-288 for 600 s to measure the dissociation rate of the complex in different conditions. To analyse the affinity between pro- and mature domains of activin A, biosensors immobilized with pro-domains at 25 µg ml$^{-1}$ were dipped into the solution containing mature activin A at different concentrations for 300 s as the association phase and then moved to the blank buffer for 300 s as the dissociation phase. Similar experiments were performed using biosensors immobilized with pro-activin A complex at 50 µg ml$^{-1}$. Before dipping into activin A solutions, the immobilized biosensors were treated with 16 µM FST-288 to scavenge all the mature activin A from the pro-activin A complex on the biosensors. After 90 s of baseline stabilization, these treated biosensors were dipped into solutions containing mature activin A for 600 s and then into blank buffer for 600 s.

The interactions between FST-288 and activin A in different protein forms were performed using streptavidin (SA) biosensors with the biotinylated FST-288 immobilized on the biosensors. To produce the biotinylated FST-288, 100 µg of Avi-tagged FST-288 was incubated with 15 units of BirA enzyme in PBS buffer containing 10 mM ATP, 10 mM magnesium acetate, 50 mM D-biotin for 1 h at 30 °C before the protein was purified by reverse phase chromatography and vacuum-dried. The biotinylated FST-288 was then suspended in the kinetic buffer (PBS, 0.1% bovine serum albumin and 0.02% Tween-20) to the concentration of 50 µg ml$^{-1}$. Activin A, pro-activin A complex and pro-activin A precursor were diluted in the kinetic buffer to the concentrations as shown in the experiments. Before the experiment, the streptavidin sensors were regenerated 5 s in 10 mM glycine pH 1.7 for three times with neutralization in the kinetic buffer in between. After measurement of 60-s baseline, the biotinylated FST-288 was loaded onto the tips for 600 s after which the rest of the streptavidin-binding sites were blocked with 100 µg ml$^{-1}$ biocytin (biotin–lysine conjugate) for 300 s. After further 120 s of baseline stabilization, the association phase of the binding was performed for 600 s and then followed by dissociation for 600 s.

The data were analysed and fitted by Prism 6 software (GraphPad) using kinetic model as follows.

Dissociation model:

$$R = R_0 e^{-k_{off}(t-t_0)},$$

where $R$ is BLI response, $t$ is time, $R_0$ and $t_0$ are the starting response and time of dissociation phase.

Association model:

$$R = \frac{k_{on}CR_{max}}{k_{on}C + k_{off}} \left(1 - e^{-(k_{on}C + k_{off})t}\right),$$

where $C$ is the concentration of analyte, $R_{max}$ is the maximum response at the equilibrium with the maximum concentration of analyte.

**Crystallization and structure determination.** Engineered pro-activin A precursor and complex with truncation of the Lys-rich region (K259-D282) were concentrated to 15 mg ml$^{-1}$ and used to set up 96-well crystallization plate using Mosquito Crystal (TTP Labtech) crystallization robot. Crystals of engineered pro-activin A precursor were obtained from sitting drops consisting of 200 nl of protein and 200 nl of crystallization solution. Crystallographic data were collected at beamline I04 at Diamond Light Source synchrotron using a mini-kappa goniometer to improve the separation of diffraction spots on the detector. Multi-wavelength anomalous dispersion data were collected under three wave-lengths (0.9793, 0.9795 and 0.9720 Å) from the crystal of selenomethionine-labelled pro-activin A precursor with truncation of Lys259-Asp282, grown in 11.4% PEG 400, 12.9% PEG 2000MME and 100 mM HEPES pH 7.1 at the protein concentration of 15 mg ml$^{-1}$. A single-wavelength anomalous dispersion data set was collected at a wavelength of 0.9793 Å from the crystal of the same protein grown in 25% PEG 1000 and 100 mM MES pH 6.5. A native data set was collected from a crystal of cleaved pro-activin A complex with truncation of K259-D282, grown in 20% PEG 3350 and 200 mM calcium chloride.

Raw data were indexed and integrated using XDS software[43] and scaled using Aimless in CCP4 suite[44]. Experimental phasing was performed using autoSHARP[45] and molecular replacement was performed using Phaser in CCP4 suite[46]. The model was manually corrected using Coot 0.8.1 (ref. 47) and refined using Refmac5 (ref. 48) and Phenix[49]. The dihedral angles of 97.6% of all amino-acid residues in the structure of unprocessed pro-activin A precursor are in the favoured region and none of the residues in the non-allowed region. The dihedral angles of 97.1% of all amino-acid residues in the structure of pro-mature activin A complex are in the favoured region and none in the non-allowed region. All the data collection, data reduction, structure determination and refinement statistics are shown in Table 1.

**Data availability.** Coordinates and structure factors have been submitted to the Protein Data Bank under accession numbers 5HLY for the unprocessed pro-activin A precursor and 5HLZ for the pro-mature activin A complex. All additional experimental data are available from the corresponding author on request.

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

## Acknowledgements

We are grateful for the member of the Hyvonen research group for useful discussion and comments on the manuscript. Mrs Katharina Ravn is thanked for the preparation of follistatin and mature activin A used in this study. We thank Dr Trevor Littlewood for the generous gift of HEK293T cells. We thank Drs Katherine Stott and Dimitri Chirgadze for providing access to and support in the Biophysical and X-ray crystallographic facilities at the Department of Biochemistry. We are grateful for the access and support at beamlines I04 and I24 at Diamond Light Source at Harwell, UK (proposal MX9007 and MX9537) and to beam lines ID29 at the European Synchrotron Radiation Facility in Grenoble, France, with data obtained at these facilities contributing to the results presented here. We thank Ansh Labs LLC, Houston, TX for the generous gift of the anti-FST antibody. Cambridge Trust and China Scholarship Council funded the PhD studentship of X.W. for this work.

## Author contributions

X.W. performed most of the experiments with G.F. contributing to the structure determination and refinement. M.H., together with X.W., conceived the study and analysed the results. All authors contributed to the preparation of the manuscript. There are no competing financial interests.

## Additional information

**Competing financial interests:** The authors declare no competing financial interests.

