## [Peer Review File · Nature Communications]

Reviewer #1 (Remarks to the Author):

The manuscript by Wang and co-workers presents the structure of proActivinA, both as the pro-complex in which the linkage between the pro- and mature-domains has been cleaved, or in its precursor form in which the linkage has not been cleaved. Though previous studies have reported the structures of the proTGFb1 and proBMP-9 complexes, these new structures (and accompanying biochemical data) are nonetheless very important as a) the prodomains have vital roles regulating signaling activity, yet are poorly understood at the molecular level, and b) the pro-domains of TGF-beta superfamily signaling proteins are much more variable in sequence (and presumably structure) than the corresponding mature domains.

There are several important new insights that emerge from the structures of the proActA precursor and proActA complex. The first is the revelation of a new architecture not previously observed in which the inhibitory N-terminal domain (a1 and a2 helices and intervening latency lasso) bind to the mature domain from the same monomer (not the opposing monomer as in the proTGFb1 and proBMP9 structures). This is important for two reasons. The first is that it provides a simpler mechanism for the assembly of the precursor - for the architecture observed in the proTGFb1 complex, the dimerization must occur as the inhibitory N-terminal domain folds around the mature domain of another monomer, while in the proActA complex, the dimerization can occur after each monomer has fully folded (as shown in Supplementary Fig. 8). The second important point is that this new architecture (which is clearly evident from density corresponding to a-helix3 that connect the inhibitory N-terminal domain and the arm domain) highlights the possibility that the previously published proTGF-b1 structure may in fact be in error - this is a very real possibility, because in the structure of proTGFb1, there was no traceable electron density in the hinge region between the N-terminal inhibitory domain and the arm domain (residue 63-69) and its possible (based on comparable distances) that the N-terminal domain connects not to the arm domain on the "same side" of the pro complex, but on the opposite side (as in the proActA complex).

The other new finding from this study is that it shows how follistatin, which is known to bind and regulate activin, can remove activin from the grip of its prodomain. This has important implications for the delivery of activin A, though the implications of this were not further investigated/discussed in this report. The one aspect of the structure that was discussed, but not studied, was whether the signaling receptors, particularly ActRIIB which binds mature ActA with near nanomolar affinity, might also be capable of removing activin from the grip of its prodomain. This is surely relevant as the relative ability of follistatin and the signaling receptors to bind the mature signaling protein certainly has important implications for function.

The overall quality of the data presented is very high and the presentation of the results is both carefully done and concise. This report significantly advances current understanding of pro-domain structure and regulation of activity, and aside from the comments about the relative ability of

follistatin and the signaling receptors to release the mature signaling protein from its prodomain, there are no major weaknesses or flaws in this study.

Minor comments related to the contents of text and figures follow:

1. The authors should indicate what the third peak in the reverse phase chromatogram of the purified proActA complex shown in Fig. 1b corresponds to (is this some uncleaved precursor?).
2. Although the connection between the arm domain and the beginning of the mature domain seems fairly unambiguous from the structure shown in Fig. 2a, the authors should nonetheless directly state this in the text (since this is another possible place for swapping, given the gap in the electron density in this region).
3. The cartoons shown in Fig. 3a-c should be modified to explicitly show the connection between the arm and mature domains (in the way it is shown, its not obvious which mature monomer belongs to which prodomain).
4. It is not really correct to say that the "helical part in the wrist" (first sentence, Page 8). It would be better to say "palm helix" or something similar.
5. It is stated that "a remarkable, and so far unique feature" of ActA in comparison with other TGF-beta superfamily proteins is "the significant variation that dimeric activin A shows in its inter-protomer conformation". It is unclear, however, why the authors stated this as TGF-b3 has also been reported to have significant variation of its inter-protomer conformation.
6. The sentence that begins "As mentioned earlier, the a-helix of the .." should be modified to specify which a-helix (for example, the "As mentioned earlier, the palm a-helix ...").
7. In the middle paragraph on Page 11, the authors should indicate the reported Kd value of Fst-288 for ActA (as relevant background information).

8. The sentence "It thus suggests that FST-288 can inhibit activin A even when it is complexed with its pro-domain" on page 13 should be modified since this might be interpreted to mean that a ternary complex can be formed between Fst-288 and the proActA complex.

9. On page 8, the sentence that begins "The hydrophobic residues stabilize residues in the receptor binding epitope" should be modified to specificity which receptor binding epitope (type I, type II, or both).

10. Several references are made to the type I receptor binding epitope (page 14, last few lines of Fig. 4 legend); it should be emphasized in the text, the type I receptor binding epitope for activin A has not in fact been reported in the literature.

11. It seems that either the proActA or proTGFb1/proBMP9 procomplex architecture could determine the heterodimerization propensity since in both structures there are places where residues from one monomer contact the other monomer. It is unclear then, based on this, why the authors seem to suggest that the proActA architecture might facilitate heterodimerization (and implied, though not explicitly stated, perhaps better than the proTGFb1/proBMP9 architecture)?

12. Figure legend 1c should be modified so that the color green is ascribed to pro-activin A precursor.

13. It was unclear in the legend for Fig. 4b whether the proActA precursor or proActA complex was used.

Reviewer #2 (Remarks to the Author):

This manuscript describes a detailed and meticulous research program to determine the degree to which the Pro region of activin remains with the mature region, its activity, and through structural investigations, its relationship to other closely related TGFb family growth factors for which crystal structures exist for mature-pro region complexes. The authors also engineered out the furin cleavage site and replaced it with a unique protease site so that they could control when the cleavage took place, if at all. They demonstrated that uncleaved pro-mature activin A is inactive but

when cleaved, the complex is active. They show that the complex is roughly 10-fold weaker in binding affinity compared to activin receptor or its natural inhibitor FST suggesting that these other binding agents knock the pro-region off the mature protein to allow interaction with other proteins. The authors then provide detailed crystal analyses of the complex and compare these results to related molecules and then explore how follistatin binds to the complex versus the mature protein.

The research is well done and described, although a number of places could use some editing for correct English usage. While the results completely support the conclusions, it is difficult to ascertain the novelty of the findings. The requirement for separation of Pro and mature activin has been known for more than two decades (citation in manuscript). The weak association between Pro and mature domains has also been known for some time as identified in the manuscript. The relationships between activin and its binding agents and the related TGF β family ligands and their binding agents have also been known, at least to some degree. Therefore, the additional insights provided by this study are not clear, nor is it clear how it will alter the thinking or research directions in the field. It is wonderful to have such a detailed and meticulous structural basis for why already known relationships exist if only to verify their reproducibility and there may be unique binding surfaces identified in this study that will allow novel modifiers to be designed, but these are not identified by the authors so again, the novelty is difficult to assess. Overall it would seem that these data support prior observations for activin and provide a structural foundation upon which to interpret those studied, but by itself, does not change the field in a novel or significant way.

Referee 1.

1. The authors should indicate what the third peak in the reverse phase chromatogram of the purified proActA complex shown in Fig. 1b corresponds to (is this some uncleaved precursor?).

We apologised for this omission – the third peak is GST-fused HRV 3C protease. We have now stated this in the legend of Fig.1: “The third peak corresponds to the GST-tagged HRV 3C protease”. We have also added the corresponding SDS-PAGE lane to the figure and labelled the protein/band as appropriate.

2. Although the connection between the arm domain and the beginning of the mature domain seems fairly unambiguous from the structure shown in Fig. 2a, the authors should nonetheless directly state this in the text (since this is another possible place for swapping, given the gap in the electron density in this region).

We have clarified this in the text:

“The linker between the pro-domain and the mature domain, with the protease cleavage site, is visible neither in the precursor form nor in the pro-mature complex. Given the positioning and distance (~16 Å) between the end of the pro-domain and the start of the mature domain, the connectivity shown in Fig. 2a is the only feasible one; the alternative connectivity would require the linker to wrap around the complex and even as a linear distance (36 Å) this is too long for the missing ten residues to span.”

3. The cartoons shown in Fig. 3a-c should be modified to explicitly show the connection between the arm and mature domains (in the way it is shown, its not obvious which mature monomer belongs to which prodomain).

We have modified the Fig. 3a-c to show the connectivity between the pro-domain and the mature domain with dotted lines and changed the second protomer to surface representation to make the different connectivity as a whole more clear. Hopefully this is now more unambiguous.

4. It is not really correct to say that the "helical part in the wrist" (first sentence, Page 8). It would be better to say "palm helix" or something similar.

We have changed this to “palm helix”, as suggested.

5. It is stated that "a remarkable, and so far unique feature" of ActA in comparison with other TGF-beta superfamily proteins is "the significant variation that dimeric activin A shows in its inter-protomer conformation". It is unclear, however, why the authors stated this as TGF-b3 has also been reported to have significant variation of its inter-protomer conformation.

We have revised this to clarify that "... mature activin A dimer has shown significant variation in its inter-protomer conformation, more than any other TGF- β -like ligand" and removed the statement regarding uniqueness of this. We further clarify that "TGF- β 3 has also been reported to show conformational changes when it is complexed with receptors³⁶, but to a much lesser extent (115°-125°) compared to activin A (50°-108°)."

6. *The sentence that begins "As mentioned earlier, the α -helix of the .." should be modified to specify which α -helix (for example, the "As mentioned earlier, the palm α -helix ...".*

We have corrected this "*palm α -helix*".

7. *In the middle paragraph on Page 11, the authors should indicate the reported Kd value of Fst-288 for ActA (as relevant background information).*

We have added that "FST-288 has been reported to be more efficient in the inhibition of activin A signalling with a Kd of 46.5 ± 0.37 pM³⁸ and we have therefore used this shorter isoform in our studies.", along with an appropriate literature reference.

8. *The sentence "It thus suggests that FST-288 can inhibit activin A even when it is complexed with its pro-domain" on page 13 should be modified since this might be interpreted to mean that a ternary complex can be formed between Fst-288 and the proActA complex.*

We have revised this sentence to remove the ambiguity:

"This suggests that follistatin can exert its inhibitory effect on activin A even when the mature domain is complexed with its pro-domain, by dissociating this complex and binding to the mature dimer. "

9. *On page 8, the sentence that begins "The hydrophobic residues stabilize residues in the receptor binding epitope" should be modified to specificity which receptor binding epitope (type I, type II, or both).*

We have specified that this refers to type II receptor binding site and have added reference to Figure 3d where this is shown.

10. *Several references are made to the type I receptor binding epitope (page 14, last few lines of Fig. 4 legend); it should be emphasized in the text, the type I receptor binding epitope for activin A has not in fact been reported in the literature.*

We have revised the text and refer now to "putative type I receptor binding surface".

11. *It seems that either the proActA or proTGF β 1/proBMP9 procomplex architecture could determine the heterodimerization propensity since in both structures there are places where residues from one monomer contact the other monomer. It is unclear then, based on this, why the authors seem to suggest that the proActA architecture might facilitate heterodimerization (and implied, though not explicitly stated, perhaps better than the proTGF β 1/proBMP9 architecture)?*

We agree with the referee that our statement with respect to heterodimerisation could be

improved, and it is better phrased in the general context of dimerisation. We have therefore revised the text:

“This mechanism would also be well posed to facilitate dimerisation of the growth factors, as individual chains are able to fold more completely and obtain a conformation that is compatible with dimerisation. The connectivity and interactions between the pro- and mature domains could determine dimerisation propensity as only mutually compatible monomeric pro-forms could interact with each other in either homo- or heterodimeric manner.”

12. Figure legend 1c should be modified so that the color green is ascribed to pro-activin A precursor.

This has been corrected.

13. It was unclear in the legend for Fig. 4b whether the proActA precursor or proActA complex was used.

Pro-Activin A precursor has been used. The legend has been corrected and PDB accession codes included for both type II (1s4y) and follistatin complexes (2b0u).

Referee 2.

The research is well done and described, although a number of places could use some editing for correct English usage

We have revised the text throughout and hopefully corrected the text to referee's satisfaction.

Due to editorial request to limit the main text word count to 5000, we have edited the text throughout to fit within this limit.